# Osteochondral Lesions of the Ankle Treated with Bone Marrow Concentrate with Hyaluronan and Fibrin: A Single-Centre Study

**DOI:** 10.3390/cells11040629

**Published:** 2022-02-11

**Authors:** Sameera Abas, Jan Herman Kuiper, Sally Roberts, Helen McCarthy, Mike Williams, Andrew Bing, Bernhard Tins, Nilesh Makwana

**Affiliations:** 1Department of Foot and Ankle Surgery, Robert Jones and Agnes Hunt Orthopaedic Hospital NHS Foundation Trust, Oswestry SY10 7AG, UK; jan.kuiper@nhs.net (J.H.K.); sally.roberts4@nhs.net (S.R.); helen.mccarthy@nhs.net (H.M.); mike.williams17@nhs.net (M.W.); andrew.bing@nhs.net (A.B.); btins@nhs.net (B.T.); 2School of Pharmacy and Bioengineering, Keele University, Staffordshire, Newcastle upon Tyne ST5 5BG, UK

**Keywords:** articular cartilage, bone marrow concentrate, osteochondral defect, talus, ankle

## Abstract

Osteochondral defects of the ankle (OCD) are being increasingly identified as a clinically significant consequence of injury to the ankle, with the potential to lead to osteoarthritis if left untreated. The aim of this retrospective cohort study was to evaluate a single-stage treatment of OCD, based on bone marrow aspirate (BMA) centrifuged to produce bone marrow concentrate (BMC). In a dual syringe, the concentrate was mixed with thrombin in one syringe, whereas hyaluronan and fibrinogen were mixed in a second syringe. The two mixtures were then injected and combined into the prepared defect. Clinical outcome and quality of life scores (MOXFQ and EQ-5D) were collected at baseline and yearly thereafter. Multilevel models were used to analyse the pattern of scores over time. Ninety-four patients were treated between 2015 and 2020. The means of each of the three components of the MOXFQ significantly improved between baseline and 1 year (*p* < 0.001 for each component), with no further change from year 1 to year 3. The EQ-5D index also improved significantly from baseline to 1 year, with no evidence for further change. Our results strongly indicate that this BMC treatment is safe for, and well tolerated by, patients with OCD of the ankle as both primary treatment and those who have failed primary treatment. This technique provides a safe, efficacious alternative to currently employed cartilage repair techniques, with favourable outcomes and a low complication rate at 36 months.

## 1. Introduction

An osteochondral defect (OCD) is broadly defined as a defect involving both the articular cartilage and adjacent subchondral bone [1]. However, there is some debate about the true definition of osteochondral defects, with other authors expanding the definition of osteochondral lesions as a lesion of any origin involving the articular cartilage and/or adjacent subchondral bone, thus expanding the definition to include lesions limited to cartilage, limited to bone, and affecting both [2]. In the ankle (specifically the talus and plafond of the distal tibia), both traumatic and non-traumatic etiologies have been described. The most reported cause of OCDs of the ankle is trauma, specifically recurrent ankle sprains. Berndt and Harty proposed that lateral injuries occur with inversion and dorsiflexion of the ankle, while posteromedial injuries are the consequence of ankle plantar flexion and inversion injury [3], a notion which was supported in subsequent studies [4]. Osteochondral lesions of the ankle are increasingly being recognized as a clinical problem, as the likely consequence is osteoarthritis of the ankle if left untreated, with subsequent significant loss of function for the patient. The prevalence of osteochondral lesions of the talus is 0.002 per 1000 persons and they occur in 6.5 out of 100 ankle sprains, although reports of their prevalence in ankle injuries have been as high as 50% of acute ankle sprains and fractures [1]. In a recent meta-analysis of 181 studies, the incidence of ankle sprain injuries was 13.6 per 1000 exposures in females and 6.94 per 1000 exposures in males [5]. These injuries are therefore more common than had previously been recognized.

The results with non-surgical treatments have been suboptimal [6,7]. Surgical treatment can be broadly characterized into traditional debridement and excision of loose bodies or damaged cartilage, bone marrow stimulation techniques, cell-based repair techniques and use of biological agents. Surgical options include excision, excision and debridement of damaged cartilage, microfracture (MF), autologous or allograft osteochondral implantation (OAT) and autologous chondrocyte implantation (ACI). More recent techniques include the use of particulate juvenile articular cartilage (PJA), platelet-rich plasma (PRP), bone marrow concentrate (BMC) and mesenchymal stem cells [8]. Particulate juvenile articular cartilage therapy (PJA) involves the harvesting of small-particle or minced articular cartilage from juvenile allograft donors. This allograft has been demonstrated to have a higher proportion of pluripotent chondrocytes with the ability to form new cartilage similar to hyaline cartilage, as compared to adult cartilage grafts [2], however, comparison of this technique with traditional microfracture did not demonstrate any significant benefit [9]. MF is currently considered the “Gold Standard” for primary treatment of lesions <1.5 cm^2^ due to its relatively low cost, ease of use and good short to medium term outcomes in up to 85% of cases [10,11]. Some studies have shown good to excellent short to medium term results in over 70% of cases in the talus [2,6,12]. However, other studies report poor outcomes, with low quality fibrocartilage reparative tissue (containing mainly type 1 collagen rather than the type II collagen typical of hyaline cartilage) and deteriorating outcomes at longer term follow-up, going up to six years [13,14]. Even at two year follow-up, poor radiological and deteriorating functional results have been seen [15]. In addition, second look arthroscopy confirms incomplete healing in 36% of lesions, with inferior quality of the repair tissue at an average of 3.6 years [16]. Failed primary treatment with MF can be treated by using osteochondral autograft transfer (OAT). This involves taking osteochondral plugs from the knee or talus and transplanting these into the OCD through a medial or lateral malleolar osteotomy. A single or multiple plug (mosaicplasty) can be used with good short to medium term results [17,18]. However, concerns exist regarding donor site morbidity and graft integration with surrounding bone and cartilage as well as the need for an osteotomy [19].

Autologous chondrocyte implantation (ACI) is a two-stage procedure where hyaline cartilage is harvested from the anterior aspect of the talus or a lesser-weight bearing surface in the knee such as the intercondylar notch or trochlea, from which chondrocytes (cartilage cells) are isolated and cultured in an accredited good manufacturing process (GMP) facility. The cells are then delivered in a second procedure into the OCD and covered with either a periosteal patch or a collagen membrane [20]. The chondrocytes can also be first integrated onto a collagen membrane (matrix-induced ACI (MACI), and then placed directly into the defect. Whilst good results are reported [21,22,23], the treatment is expensive and NICE have not approved either of these cell therapy approaches for use in the ankle in the UK. Three systematic reviews [6,12,22] and one Cochrane review [23] have failed to show superiority of any of these treatments for OCDs of the ankle and advise that better quality data is required.

Mesenchymal stromal or stem cells (MSCs) have been studied for over 50 years [24], particularly those isolated from bone marrow, and there has been a growing interest in the use of MSCs for the repair of cartilage defects, as freshly isolated bone marrow aspirate (BMA), more concentrated mononuclear cells (MNC), and also culture-expanded MSCs in a GMP facility [8]. Bone marrow concentrate MSC (BMC), together with hyaluronan (also known as hyaluronic acid, HA) and fibrin gel, has been used successfully in the knee [25]. Studies have demonstrated that hyaluronan maintains viability of cultured chondrocytes, thereby facilitating them to generate cartilage [26,27], leading to the production of tissue that resembles hyaline cartilage [28]. The use of fibrinogen has been shown to potentiate the generation of cartilage by chondrocytes in vitro; it is also viscous enough for easy use as an injectable carrier at the defect site [29] and has hemostatic properties. Shetty (2014) reported on 30 patients with osteochondral lesions in the knee with ICRS grade III/IV who were treated with a combination of BMC, HA and fibrin [25]. The results showed a significant clinical improvement, with morphological changes on the MRI showing good cartilage defect repair. BMC alone has also been used in the ankle for OCD. Murphy et al (2019) reported their outcomes comparing BMC to MF in 49 and 52 patients respectively and found the technique to be safe and effective with a lower revision rate compared to MF [30].

The purpose of this study was to review a single-center experience of using BMC in combination with hyaluronan and fibrin for the treatment of primary and non-primary OCDs of the ankle. The definition of OCDs in this study mirrors that used by our colleagues to describe OCDs in the knee, i.e. ICRS grade III/IV [25]. We present our experience of a single-stage technique that can be considered a hybrid of cell-based repair and biologic agent technique.

## 2. Materials and Methods

This publication adhered to the Minimum Information for Studies Evaluating Biologics in Orthopedics (MIBO) reporting guideline for Mesenchymal stem cells (Table A1) [31], and Strengthening the Reporting of Observational Studies in Epidemiology (STROBE) reporting guideline for cohort studies (Table A2) [32].

### 2.1. Patient Selection

This was a single-center retrospective review of data collected prospectively between March 2015 and March 2020 from all our patients with osteochondral defects of the ankle undergoing treatment with BMC combined with hyaluronan and fibrin (Table 1). Our inclusion criteria were: (1) skeletally mature (aged 15 years and above), (2) osteochondral defects of the ankle (talus or tibial plafond) as confirmed via imaging or arthroscopically, (3) symptoms persisting for longer than six months, and (4) failed primary conservative care or primary surgical treatment. Exclusion criteria were: (1) established osteoarthritis (Kellgren-Lawrence Grade 4), (2) inflammatory arthritis, (3) gross malalignment of the ankle, and (4) “kissing lesions” i.e., concurrent lesions of both the talus and the tibial plafond.

### 2.2. Bone Marrow Aspirate Concentrate (BMC)

The technique used for preparing the BMC to be injected into the osteochondral defect has been described previously [25]. This involves harvesting 35 mL of bone marrow aspirate from the patient (either from the anterior or posterior iliac spine of the pelvis; the area was marked, cleaned with chlorhexidine or betadine preparation and draped) which was mixed with ACDA (an anticoagulant of sodium citrate dehydrate, glucose, and citric acid; Fresenius KABI, Bad Homburg, Germany). A bone marrow aspirate concentrate (BMC) was produced via centrifugation of the aspirate in the operating theatre, containing mononuclear cells. This was not evaluated microscopically. 0.8 mL of BMC was then combined with 0.2 mL thrombin (Tisseel^®^, Baxter, Thetford, UK) and calcium chloride, and loaded into one barrel of a dual Y-shaped syringe. A mixture of 0.2 mL HA (10 mg/mL of high molecular weight HA, High HyalPLus manufactured by Humedix, Republic of Korea) and 0.8 mL fibrinogen and aprotinin (Tisseel^®^, Baxter, Thetford, UK, was loaded into the other barrel of the Y-shaped syringe, according to the manufacturer’s instructions (Regen Global UK, CCR Kit^®^). The combined volume of the two barrels of the dual syringe was 2 mL. The contents of the dual syringe were deployed to the prepared osteochondral defect, which had been debrided back to cartilage with a macroscopically healthy appearance; this was done either arthroscopically or in an open procedure. Of the final 2 mL volume created using this technique, the volume deployed to treat each OCD was as much as was needed to fill the defect. This varied according to the size of each individual OCD.

### 2.3. Surgical Technique

For the arthroscopy or open procedure to be performed, the patient was positioned supine with the affected leg on a knee bolster and underwent either a general or spinal anesthetic. An ankle stirrup was used to apply traction, and a high thigh tourniquet was applied and inflated prior to arthroscopy. The defect was debrided arthroscopically in most cases; deep or posterior lesions in the ankle joint required an open or malleolar osteotomy for access. Cysts were debrided and bone grafted using local autologous bone from the tibial metaphyseal area. Once the lesion was dried, the gel complex was then applied to the defect. MF was performed where the subchondral bone was intact. The ankle was then taken off traction, (or, in the case of osteotomy, this was reduced back), and then taken through its range of movement with simulated weight bearing. The lesion was then re-inspected to ensure that the gel complex was stable and had not displaced. Wounds were closed with 3/0 nylon.

### 2.4. Post-Operative Protocol

Post-operatively, patients were told not to bear any weight on the affected leg for two weeks and were given crutches. They were then commenced on a structured physiotherapy regime, starting with introducing partial weight bearing back to the leg and then progressing on to return to full weight bearing over the subsequent two weeks. Those patients who underwent osteotomy were kept in a plaster-of-Paris cast or an Aircast boot for six weeks, with range of movement exercises commencing at week 2 post-operatively if the osteotomy remained stable. The progression from partial to full weight bearing was commenced at six weeks, while preventing high-impact loading for six months.

### 2.5. Outcome Measures

Manchester-Oxford Foot and Ankle Questionnaire (MOXFQ, [33]) and EQ-5D-5L scores were taken pre-operatively, and at 3, 6, 12, 24 and 36 months. The MOXFQ is a functional foot and ankle score consisting of three sub-scales (pain, walking/standing and social interaction) and a summary (or MOXFQ-Index) score; each have a range of 0 to 100 (100 being the worst). The EQ-5D-5L is a standardized way of measuring health status developed by the EuroQol Group in order to provide a simple, generic health measurement for clinical and economic appraisal [34]. Based on the UK value set, the EQ-5D-5L ranges from −0.594 to 1, with 1 representing perfect health, 0 representing death, and values below 0 representing health states worse than death.

Post-operative MRI scanning was not routinely carried out for all patients in the cohort. However, in our cohort, 40 patients underwent MRI scanning post-operatively. We subsequently used data from the scans to calculate Magnetic Resonance Observation of Cartilage Repair Tissue (MOCART) scores within 6 months of performing the scans; the MOCART is a scoring system which has been validated for examining the morphological features of cartilage defects [35].

### 2.6. Statistical Analysis

QQ-plots were used to decide if continuous baseline variables were normally distributed. If distribution is non-normal, values were summarized using medians and quartiles. Linearly segmented multilevel models were used to analyze the pattern of mean outcome scores (MOXFQ and EQ-5D) over time. Multilevel models were chosen to correctly handle any missing outcome data. In these models, we assumed there would be an early post-operative first phase during which the scores would change rapidly, followed by a second phase comprising the remainder of the follow-up period during which scores would change slowly, in line with other outcome studies on patients recovering from joint surgery [36,37]. The time of the transition between the two segments or phases can differ between different outcome types [37]. We therefore determined optimally fitted transition points (changepoints) in the models for each outcome [38]. Models were fitted using random intercepts and random slopes for phase 1, random transition points, random slopes, and a random quadratic term for phase 2, with log-likelihood ratio (LR) tests being used to decide the statistical significance of the random terms. We used these models to determine mean outcomes at baseline, 1 year and 3 years, and their 95% confidence intervals. EQ-5D scores are known to show skewness and heteroskedasticity, but we reported mean values as these are used in health economics. However, we used robust (sandwich) variance estimates when determining EQ-5D results [39]. For models of the MOXFQ, QQ-plots were used to check if the residuals were distributed normally. Once these mixed models had been determined, we did further analyses to find potential baseline demographic and clinical features predicting the rise in scores during phase 1 by introducing interaction terms of baseline feature and phase 1 slope. This analysis started with full models (including all interaction terms) followed by augmented backward elimination, removing at each step the feature that most reduced the corrected Akaike Information Criterion (AICc) until either the solution with minimal AICc was found or the coefficients of each remaining feature started to deviate noticeably from the coefficients in the previous step as based on their 95% confidence interval [40]. In case of a bilateral procedure, the two ankles were analyzed independently, since their dependency has been shown to have little practical consequences on analysis results [41]. When considering previous surgery as a predictor, we compared the use of a binary (no/yes) and ternary classification (no/microfracture/other). For the MRIs, we investigated if there was a correlation between MOCART score and time since operation, and between MOCART score and concurrent MOXFQ summary index score as determined using the mixed model. For all analyses, we assumed a *p*-value below 0.05 to denote statistical significance. All statistical analyses were performed using R vs 4.0.5 (R Foundation for Statistical Computing, Vienna, Austria), using the “nlme”, “segmented”, “clubsandwich”, “emmeans” and “effects” packages. At the beginning of the study, we performed a sample size analysis. Based on the published MCID of the MOXFQ in ankle surgery patients (13 points for each of the subscales) and its SD of change (29 points at most), the required sample size to demonstrate the MCID at the *p* = 0.05 level using a 2-tailed repeated *t*-test assuming 80% power was 42 patients [42].

## 3. Results

### 3.1. Demographics

All continuous baseline variables except the time from injury, symptom onset and EQ-5D were distributed normally. Ninety-four patients had BMC treatment as either the primary treatment (62 ankles) or following a previous failed treatment (34 ankles) for osteochondral defects of the talus and tibial plafond between March 2015 and March 2020. The mean age was 37.3 years (range 15–72). The ratio of left side to right was 1:1.64. Two patients underwent bilateral surgery. Mean BMI was 29.3 (S.D. 5.6). While 70 patients had an identified mechanism of injury, 24 patients were unable to recall a specific injury or index event causing their symptoms. Defect size ranged between 0.4 and 4.0 cm^2^, with a mean area of 1.5 cm^2^, comparable to other studies examining the BMC technique [13,25]. Baseline characteristics are summarized in Table 1.

Among the 62 patients in our study who had undergone surgery prior to BMC, arthroscopy plus microfracture or arthroscopy with for instance debridement were the most common (Table 2). Twenty of the patients in the study demonstrated osteoarthritis pre-operatively. In these patients, the degree of osteoarthritis was assessed using the Kellgren-Lawrence classification on pre-operative anterior-posterior (AP) X-rays. In four patients, further supplementary CT imaging was used to confirm the presence of osteoarthritis and to assist with grading; in one patient, MRI was obtained to further assess osteoarthritis and assist with grading. In that patient, X-ray findings were normal (Kellgren-Lawrence stage 0), but osteoarthritis was demonstrated on MRI. Further breakdown of Kellgren-Lawrence grading in the 20 patients is outlined in Table 3.

### 3.2. MOXFQ Scores and EQ-5D Scores (Patient-Related Outcome Measures)

The mean follow-up time was 12 months, with a maximum of 46 months. The residuals of the MOXFQ multilevel models were normally distributed. All best-fit models had a random intercept and a random slope for phase 1, but no random slope for phase 2. For phase 2, the MOXFQ models for walking, social interaction and summary index had significant linear (*p* = 0.0015, 0.009 and 0.0034 respectively.) and quadratic components (*p* = 0.020, 0.015 and 0.0034 respectively.), whereas no evidence was found for a linear component in the model for the pain component (*p* = 0.31). Across all domains of the MOXFQ score, we observed an initial rapid reduction over time of the score compared to baseline scores, and over the follow-up period, a sustained improvement in scores (Figure 1). For all MOXFQ outcomes, the transition between the initial rapid improvement and more steady state was estimated to occur at 1.8 months. Over the 3-year follow-up period, reduction in MOXFQ scores in all domains was observed compared to baseline (Table 4 and Figure 1). The difference between baseline and 12-month MOXFQ scores across all domains was statistically significant (*p* < 0.001). However, no evidence was found for a difference between MOXFQ outcome measures at 36 months compared to those at 12 months.

Based on the model for the EQ-5D score, the transition between initial rapid rise and a steadier phase 2 occurred at 5.5 months, with no evidence for a further change during phase 2 (mean slope −0.043 per year, 95%CI −0.095 to 0.009, *p* = 0.10, Figure 2). The 12-month EQ-5D score was significantly improved compared to baseline, but no statistical evidence was found for a further change until 36 months (Table 4).

### 3.3. Predictors of Improvement in MOXFQ-Summary Index

Based on our data, the most important predictors of the initial reduction in MOXFQ summary scores (better outcome) compared to baseline were: not having had an injury, shorter time from symptom onset, no previous surgery, no signs of osteoarthritis, and a larger area of the defect (Table 5). Having had an injury, previous surgery or signs of OA each give around 8 points less improvement. The longer the symptoms, the less improvement (0.7 points per year). The larger the defect, the more improvement was observed in the patient’s MOXFQ score (around 7 points per cm^2^). Characteristics for which we did not find evidence of an effect on improvement were age, sex, BMI, affected bone (talus or tibia), defect location on bone, presence of bone oedemas, presence of concurrent cysts, or an intraoperative osteotomy with the BMC. When comparing the binary and ternary classification of previous surgery, we found no evidence that splitting the category between “yes”, “microfracture”, and “other” improved prediction (likelihood ratio test, *p* = 0.97), and we therefore kept the yes vs no split.

### 3.4. Post-Operative MRI Scan Findings

Post-operatively, 40 patients, all with a minimum of 12 months follow-up, underwent MRI scanning (median 15 months post-operatively, range 2–60 months). For 10 patients, scans were undertaken earlier than the routine 12-month follow-up. We calculated Magnetic Resonance Observation of Cartilage Repair Tissue (MOCART) scores within six months of performing the scans, a scoring system which has been validated for examining the morphological features of cartilage defects [35]. The mean MOCART score was 62 points (range 30 to 90). For every year of follow-up, we found a mean loss of 6.5 MOCART points per year (95%CI −0.7 to 13.6, *p* = 0.074). We found no evidence for a correlation between MOCART and concurrent functional outcome (r = −0.07, 95%CI −0.42 to 0.38, *p* = 0.65, Figure 3).

### 3.5. Complications

Ten patients underwent arthroscopy post-operatively due to the development of clinical symptoms such as ongoing pain and reduced range of movement. Four patients developed complex regional pain syndrome (CRPS), one patient developed a neuroma and three developed stiffness and reduced range of movement. One patient underwent subsequent total ankle arthroplasty for persistent pain and multifocal disease, and another patient underwent ankle fusion due to development of persistent pain and joint degenerative changes. We were fortunate not to lose any patients to follow-up, although one patient was discharged after six months due to their geographical relocation.

## 4. Discussion

OCD of the talus remains an important cause of continued post-traumatic ankle pain. Current treatment strategies such as conservative management (reported to be successful in up to 55–60% of cases in select population groups [6,43]), microfracture and autologous chondrocyte implantation (which regenerate cartilage of different quality to native hyaline cartilage [12,44,45,46]) are widely employed with reasonable levels of success in select patient groups. However, such measures have their limitations; in the case of microfracture, the length of time that the integrity of the cartilage regenerated remains is limited, and the quality of the cartilage produced is inferior to native hyaline cartilage. Although a 96% rate of success has been reported in athletes for microfracture and bone grafting at 2 to 8 years post-operatively [1] and systematic reviews support the high success rate for stimulation techniques [6], no studies demonstrating the long-term quality of the repair and retention of integrity exist yet. The longest follow-ups reported in literature are approximately 5–10 years [12,47,48,49]. A study of 59 patients’ ankles treated with ACI in our center showed that 69% of patients were ‘pleased’ or ‘very pleased’ at a mean follow-up point of 5.1 years (2.3–14.6 years), but here the surgery was more complex and required two procedures [50,51].

The potential for pluripotent bone marrow MSCs to differentiate into osteogenic and chondrogenic cells, and hence the potential to regenerate cartilage, has long been postulated, since it was reported by Friedenstein and colleagues [24] yet this form of therapy for the treatment of osteochondral defects has only recently started becoming more prominent and promising [46,47]. We have demonstrated that BMC leads to a significant improvement in patient-reported outcomes in the first 12 months and that the improvement was sustained over the follow-up period (36 months). The initial rapid benefit is greater if the cause of injury is atraumatic, if BMC is the primary surgical treatment (with no previous procedures), if there are no signs of early osteoarthritis and if the patient has had a short duration of symptoms. We chose a standardized measure of health status questionnaire, the EQ-5D, as well as a joint specific functional outcome, the Manchester-Oxford Foot and Ankle Questionnaire (MOXFQ). Patients showed an initial improvement with respect to our selected outcome measures, the effects of which were sustained over our 36-month follow-up period. For those patients who underwent MRI scanning post-operatively, we correlated MRI findings with their clinical picture using the 3D-MOCART score.

Our study’s strengths include a long follow-up period, which was observed in a large cohort of patients undergoing BMC for primary and non-primary OCD of the ankle (36 months), low re-operation rate and zero follow-up loss. Our reported re-operation rate (10.1%) is lower than that of our colleagues who have previously examined BMC in the ankle and reported a 12.2% re-operation rate compared to 28.8% for microfracture [28,30]. Other studies have also reported higher complication rates in traditional microfracture alone as compared to microfracture with adjuvant BMC use [48,49].

The data presented here is from a series of patients treated in a single specialist center for foot and ankle surgery and, as such, has limitations associated with a single-center cohort study. In addition, this was an observational study carried out retrospectively, with no specified minimum follow-up time, which was further limited by not having a comparison group, such as HA or BMC alone; hence it is not possible to be sure if the major contributor to the clinical improvement following treatment is due to the BMC or HA per se. Our choice to include all patients treated up to 31 March 2020 has the obvious disadvantage that not all patients reached the 36-month follow-up point. However, our statistical method was appropriate to handle such differences in follow-up timescales, and therefore our conclusions remain valid.

We did not examine one specific patient group e.g. athletes, or make a comparison between BMC patients and other patients treated primarily with microfracture or ACI. Recently published data, however, suggests that post-operative MRIs in patients undergoing BMC treatment yields superior improvement to radiological appearance as compared to microfracture alone [48,49].

We did not examine the histology of the patients we treated post-operatively, nor did we routinely assess integration of the BMC treatment with native cartilage via arthroscopy. Routine post-operative MRIs were not carried out in every patient; however, we were able to obtain MRI scans for 40 patients in our cohort. These assessments are not currently standard practice following BMC treatment and such measures are only employed if clinically indicated (for example to investigate a source of post-operative pain). Of the patients with pre-existing osteoarthritis (Table 3), we cannot report on any worsening in the severity of this, as post-operative imaging was not routinely performed. We also did not formally analyze the MSC content of the final mixture that was used on the individual OCD for each patient by examining the contents of each syringe microscopically before deployment. Although approximate numbers of cells could be construed based on previous studies, further studies are required to ascertain the number of cells obtained in the final volume via the BMC preparation technique that we have utilized here.

## 5. Conclusions

BMC with hyaluronan and fibrin is a safe treatment in patients undergoing primary treatment for OCDs of the ankle, and importantly also for those whose primary treatment has failed. We have demonstrated in our cohort that this single-procedure technique is well-tolerated by patients and avoids the two surgical procedures required for ACI. It can be used with reasonable effectiveness in patients with osteochondral defects of the ankle including those who have cysts in the underlying bone. Our results suggest that the single-step technique using BMC is a good treatment option for cartilage repair in the ankle, with associated improved functional outcome scores.

The clinical outcome at 36 months remains favourable with a low complication rate and patients were generally satisfied with the procedure. To further assess the effectiveness of this technique, longer follow-up and ideally a multicenter, randomized, controlled trial is required.

## Figures and Tables

**Figure 1 cells-11-00629-f001:**
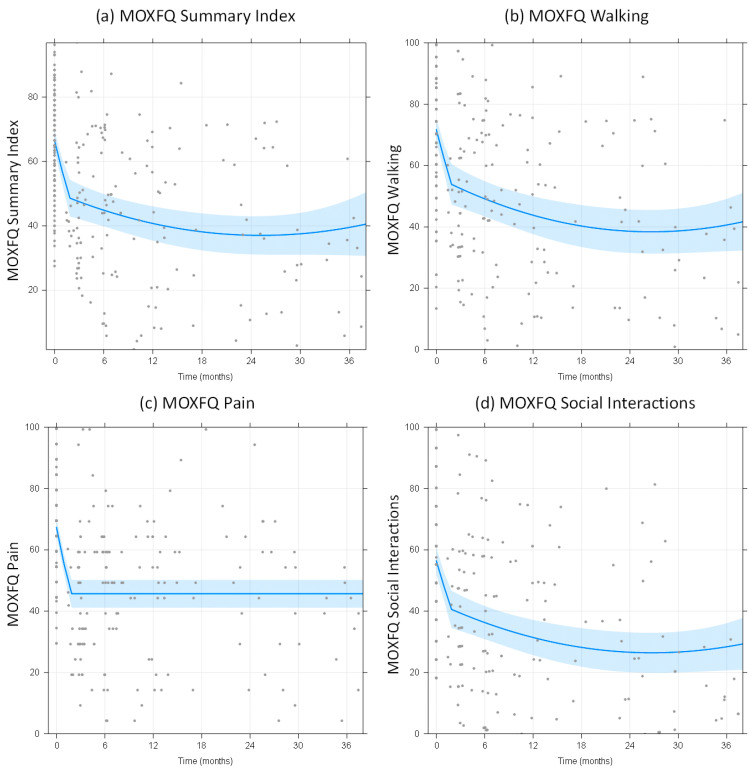
Mean MOXFQ for (**a**) summary index and sub-components of MOXFQ: (**b**) walking score (**c**) pain score (**d**) social interaction score over time, showing an initial rapid reduction compared to baseline scores with a sustained improvement over the follow-up period. In all 4 figures, the light shaded area represents the 95% CI band, and the grey dots represent all individual datapoints.

**Figure 2 cells-11-00629-f002:**
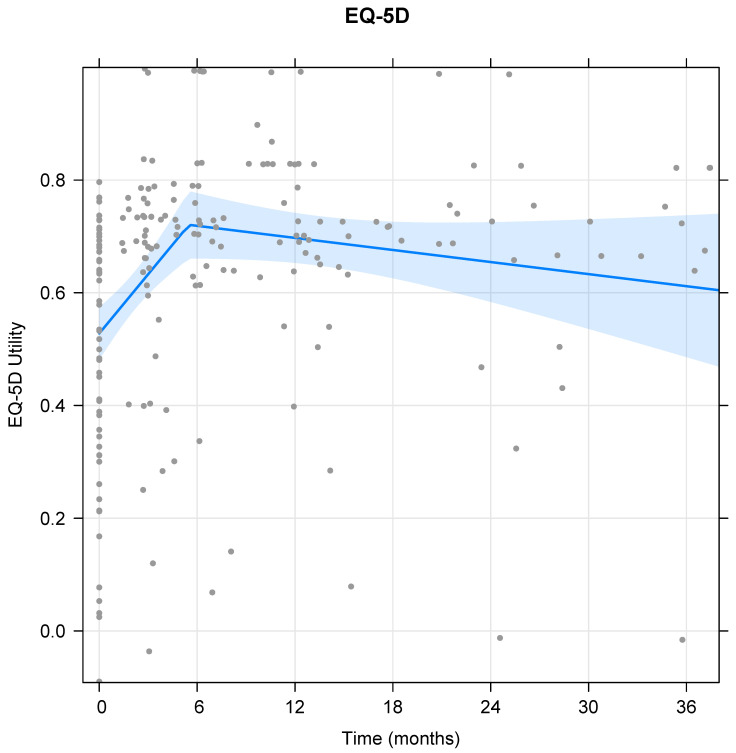
Mean EQ-5D-5L utility value over time. The light shaded area represents the 95% CI band, and the grey dots represent all individual datapoints.

**Figure 3 cells-11-00629-f003:**
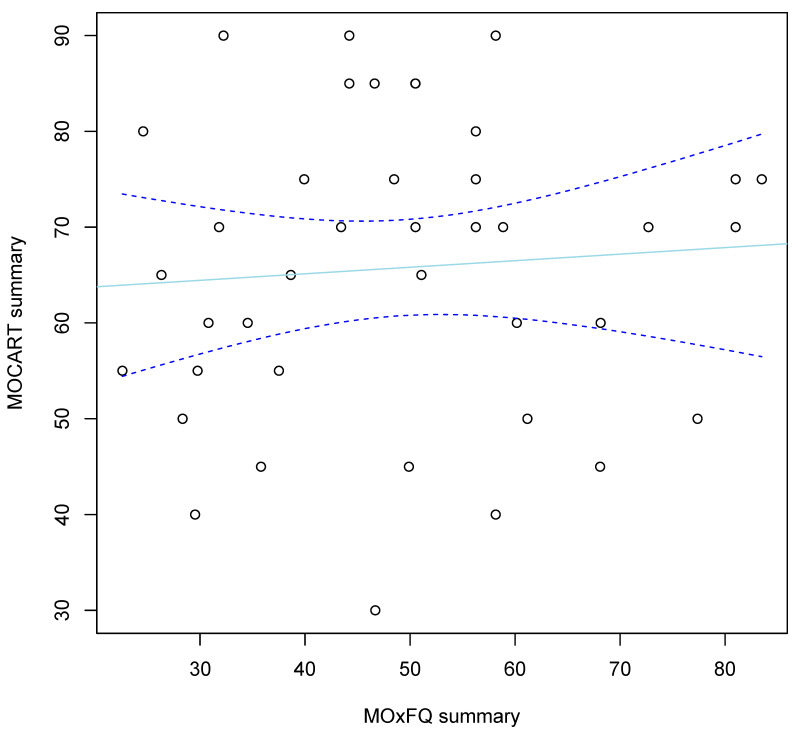
Plot of MOCART scores vs concurrent MOXFQ summary index scores. No evidence was found of a correlation (r = −0.25, 95%CI −0.62 to 0.20). Best-fit line: solid light-blue, 95% CIs: dashed dark-blue lines.

**Table 1 cells-11-00629-t001:** Baseline demographic and clinical characteristics.

Parameter	Level	Mean (SD), Median [Range] or *n* (%)
Number of patients (ankles)		94 (96)
Age (mean (SD))		37.3 (14.4)
Sex(%)	M	51 (54)
	F	43 (45)
BMI (mean (SD))		29.3 (5.6)
Bone affected (%)	Talus	83 (88)
	Both Talus and Tibia	8 (8)
	Tibia	3 (3)
Location (%)	Medial Talus	65 (76)
	Lateral Talus	16 (19)
	Both Medial and Lateral Talus	3 (4)
	Central Talus	1 (1)
Known history of injury (%)	Yes	70 (74)
	No	24 (26)
Months from symptoms onset (median [range])		66.5 [19, 372]
Injury mechanism (%)	Fall	37 (54)
	Sport	29 (41)
	Horse	2 (3)
	Road/Traffic Accident	2 (3)
Months from injury (median [range])		60 [8, 480]
Previous surgery (%)	Yes	62 (65)
	No	34 (35)
Bone oedemas (%)	Yes	75 (79)
	No	20 (21)
OA (%)	No	75 (79)
	Yes	20 (21)
Cysts (%)	Yes	63 (66)
	No	33 (34)
Area (cm^2^; mean (SD) [range])		1.5 (0.7) [0.4 to 4]
Osteotomy (%)	No	83 (88)
	Yes	13 (15)

Note: We omitted information on BMI and Cysts for 1 patient each, Months from injury and Bone oedemas for 2 patients each, and the use of an Osteotomy for 8 patients.

**Table 2 cells-11-00629-t002:** Patients that had previously been operated on: details of first previous procedure and number of patients who had undergone 1, 2 and 3 previous procedures.

Previous Surgery	*n* = 62
Arthroscopy and microfracture	31
Arthroscopy	27
Open debridement	2
Open reduction and internal fixation for fracture	2
1 × previous procedure	23
2 × previous procedures	31
3 × previous procedures	8

**Table 3 cells-11-00629-t003:** Kellgren-Lawrence Classification of 20 patients with confirmed osteoarthritis on pre-operative imaging.

Kellgren-Lawrence Classification	*n* = 20
0 (no OA)	1
1 (doubtful)	5
2 (mild)	13
3 (moderate)	1
4 (severe)	0

**Table 4 cells-11-00629-t004:** Mean outcomes following BMC for OCD.

Outcome	Baseline	12 Months	*p*-Value (vs. Baseline)	36 Months	*p*-Value (vs. 12 m)
MOXFQ					
*Summary*	66.5 (63.4 to 69.7)	40.8 (35.3 to 46.2)	<0.001	39.5 (30.7 to 48.4)	0.79
*Walking*	71.7 (67.9 to 75.5)	43.8 (37.6 to 50.0)	<0.001	40.6 (32.0 to 49.2)	0.41
*Pain*	67.3 (64.3 to 70.3)	45.6 (41.0 to 50.2)	<0.001	42.7 (35.3 to 50.1)	0.31
*Social*	56.5 (52.1 to 60.8)	31.4 (25.6 to 37.2)	<0.001	28.4 (20.6 to 36.2)	0.37
EQ-5D	0.53 (0.48 to 0.57)	0.70 (0.65 to 0.75)	<0.001	0.61 (0.52 to 0.70)	0.06

Note: all values determined using a linear mixed model and given as mean (95% confidence interval).

**Table 5 cells-11-00629-t005:** Predictors of improvement in MOXFQ summary index.

Predictor	Coefficient (95% CI)	*p*-Value
**Full model**		
Age (per year)	−0.12 (−0.65 to 0.41)	0.65
Male	−3.6 (−15.3 to 8.1)	0.54
BMI	0.6 (−0.7 to 1.9)	0.36
Known history of injury ^a^	16.3 (2.8 to 29.8)	0.017
Time from symptom onset (per year)	0.7 (−0.03 to 1.4)	0.057
Previous surgery ^a^	11.3 (−1.6 to 24.2)	0.084
Bone oedemas	−3.4 (−17.0 to 10.3)	0.63
OA ^a^	6.9 (−7.4 to 21.3)	0.34
Bone affected ^b^	-	0.42
Location ^b^	-	0.71
Defect area (per cm^2^)	−6.5 (−15.5 to 2.4)	0.15
Cysts	3.3 (−13.3 to 19.9)	0.69
Osteotomy	−5.0 (−21.1 to 11.1)	0.54
**Final model**		
Known history of injury ^a^	8.1 (−0.8 to 17.1)	0.073
Time from symptom onset (per year)	0.7 (0.1 to 1.2)	0.013
Previous surgery ^a^	7.7 (−1.4 to 16.8)	0.095
OA ^a^	7.9 (−1.3 to 17.1)	0.092
Defect area (per cm^2^)	−6.7 (−11.9 to −1.5)	0.012

Note: all values were determined using a linear mixed model. The final model was determined by sequentially removing predictors whose inclusion gave the largest increase in corrected Akaike Information Criterion (AICc) until AICc was minimised. Positive coefficient values imply that the predictor increases the score and therefore worsens functional outcome. ^a^ The reference category was “No”, i.e. no known injury history, no previous surgery or no OA. ^b^ Parameter had more than two categories, hence we only reported their *p*-values.

## Data Availability

The data presented in this study are available on request from the corresponding author. The data are not publicly available due to restrictions e.g., data protection and/or ethical restrictions.

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
