# Peer review of "Osteochondral Lesions of the Ankle Treated with Bone Marrow Concentrate with Hyaluronan and Fibrin: A Single-Centre Study"

_cells, 2022, doi:10.3390/cells11040629_

Round 1
Reviewer 1 Report
The authors studied the outcome of treating osteochondral lesions in the ankle joint with a combination of BMSCs with firbin and hyaluronan.
The strength of the study is that a long post treatment observation time of 3 years for a relatively large cohort of patients could be presented. Some small methological questions could be further addressed.
Fort he discussion:
What would be the best reference treatment for comparing the outcome (ACI?), treatment with hyaluronan alone?
Probably the fibrin was not autologous – would an approach with autologous fibrin provide benefit?
Abstract:
Which type of hyaluronan was used (molecular weight range, highly cross-linked or not, stability)?
Line 14, surplus blank
Line 36: state here which definition of „osteochondral“ was used in the present study?
Line 56: „the use of particulate juvenile articular cartilage (PJA)“ since it is novel, please explain it shortly
Line 109: remove the surplus point
Methods:
Line 111: which defect sizes were included or excluded? Which mean age of patients? Refer to table 1 here. Ankle: exactly which anatomical joint component was affected (refer to table 1)? How old were the defects before treated (refer to table 1)?
Line 123: „containing approximately 72 x 106 MSCs per millilitre“ does it mean mononuclear cells or indeed MSCs? Were the concentrates checked microscopically?
Line 123: What was the total volume of the concentrate: how many cells were within 0.8 mL?
Table 1: Add the range of defect sizes
table 2: it is not clear how many patients of the 62 received 1 previous procedure and how the listed procedures are distributed on the 62 patients.
Table 3: how about the severity of OA after treatment - did it change after treatment? Line 274
Line 249: surplus point
Line 367: „is tolerated well by patients and avoids the two-stage surgery of autologous cartilage implantation“ could the results of the present study be shortly compared with those of other studies using ACI? (discussion section)
Reviewer 2 Report
The study title is focused on the Osteochondral Lesions of the Ankle treated with Bone Marrow
Concentrate with Hyaluronan and Fibrin. This is a very interesting topic to search for the specialist. Please state what is the aim of your study. You write (14) The aim of this study was to assess effectiveness of a single stage treatment of OCD, based on bone marrow aspirate (BMA) centrifuged to produce bone marrow concentrate (BMC) and (346) your aim was to report on the safety and efficacy of the BMC technique delivered in hyaluronan and fibrin. This results that you did not examined your first (14) aim because further assess the effectiveness of this technique, longer follow-up and ideally a multicentre randomised-controlled trial are required (375). Please add all limitations of the study and do not spread it into multiple places. Please rewrite conclusions. There are more interesting data in your study to show to your readers.
Reviewer 3 Report
Manuscript: “Osteochondral Lesions of the Ankle treated with Bone Marrow Concentrate with Hyaluronan and Fibrin: A Prospective Single-Centre Study”.
The authors performed a prospective observational study aiming to evaluate osteochondral lesions treated with BMC plus HA and fibrin. The topic is important and deserve to be explored. However, the manuscript cannot be accepted in the present form.
Major comments:
- Since this is a prospective observational study, it is unclear if the authors registered it on gov. Moreover, there are specific guidelines to follow when reporting this kind of studies like STROBE guidelines.
- In general, the introduction is very long.
- In the methods, the authors should use subsections.
- I suggest to clearly report inclusion and exclusion criteria.
- In the exclusion criteria the authors reported “established osteoarthritis”. I suggest to be more precise referring to Kellgren-Lawrence grade.
- Could the authors specify the molecular weight of HA used?
- Did the authors calculate the sample size before starting the prospective study? Sample size calculation must be added at the beginning of the statistical analysis section.
- The authors stated that they enrolled patients aged 18 years and above (line 111). However, the authors reported a range between 15 and 72 years in the results (line 207).
- The authors reported that 27 patients had no injury. Could the authors clarify what did have these patients?
- It is unclear to me why defect area, which was not significant in the full model, is significant in the final model in table 5.
- It is unclear why the authors did not add MF in table 5.
- The authors should improve the study limitations in the discussion. For example, no control group was included and the patients enrolled were heterogeneous (different surgical techniques, previous surgery, injury etc).
Minor comments:
Lines 36-39, lines 42-45, lines 316-318: at least a reference should be reported.
Line 118: “has bee ndescribed” should be corrected.
Table 1: “cm2” needs to be corrected.
The authors should use consistently “MOXFQ”. Could the authors check throughout the manuscript?
Figure 1-4 are very large. I suggest to put together the 4 graphs in a single figure with 4 panels.
Line 329: “OLT” is unclear.
The authors should check the spaces between words.
Abbreviations should be used consistently throughout the manuscript.
Reviewer 4 Report
In this clinical trial, the authors demonstrated that bone marrow concentrate mixed with hyaluronan and fibrinogen could improve clinical outcomes and quality of life scores for patients with osteochondral defects of the ankle.
My major contention for the study is the lack of a comparator group treated with hyaluronan and fibrinogen per se. Thus, it is not sure the major contributor of the improvement is due to the bone marrow concentrate injected or hyaluronan per se.
Other minor comments:
-Results of Table 4: Is there comparison between 12 months and baseline?
-Table 5: Injury and having an injury, are they referring to the same parameter?
-Please provide the ethics approval code
Round 2
Reviewer 2 Report
Data corrected. Accepted in present form.
Reviewer 3 Report
The manuscript has improved and the authors have answered all my questions.
Reviewer 4 Report
Just a minor additional comment. For table 5, please add the reference group for categorical variables. For example, known history of injury (yes/no as ref?), previous surgery (yes/no as ref?) etc.
